# Low Electron Temperature Plasma Diagnosis: Revisiting Langmuir Electrostatic Probes

**Guillermo Fernando Regodón** [1], **Juan Manuel Díaz-Cabrera** [2,*], **José Ignacio Fernández Palop** [1] **and Jerónimo Ballesteros** [1]

1 Departamento de Física, Campus Universitario de Rabanales, Universidad de Córdoba,14071 Córdoba, Spain; z62rehag@uco.es (G.F.R.); fa1fepai@uco.es (J.I.F.P.); fa1bapaj@uco.es (J.B.)
2 Departamento de Ingeniería Eléctrica y Automática, Campus Universitario de Rabanales, Universidad de Córdoba, 14071 Córdoba, Spain
* Correspondence: el1dicaj@uco.es

**Abstract:** This article describes a method of measurement of the current-to-probe voltage characteristic curve of a Langmuir electrostatic probe immersed in a plasma characterized by a low electron temperature that is only one order of magnitude higher than room temperature. These plasmas are widely used in industrial processes related to surface technology, polymers, cleaning, nanostructures, etc. The measurement method complies with the strict requirements to perform representative plasma diagnosis, particularly in the ion saturation zone when the probe is polarized much more negatively that the potential of the plasma bulk surrounding the probe and allows to diagnose the plasma very quickly and locally, making it possible to better monitor and control the plasma discharge uniformity and time drift. The requirements for the Langmuir probe design, the data acquisition and data treatment are thoroughly explained and their influence on the measurement method is also described. Subsequently, the article describes different diagnostic methods of the magnitudes that characterize the plasma, based on theoretical models of that characteristic curve. Each of these methods is applied to different zones of the measured characteristic curve, the obtained results being quite similar, which guarantees the quality of the measurements. The advantages and disadvantages of each method are discussed. A series of measurements of the plasma density for different plasma conditions shows that the method is sensitive enough that the temperature of the ions needs to be taken into account in the data processing. Finally, a Virtual Instrument is included in the LabView environment that performs the diagnosis process with sufficient speed and precision, which allows the scientist to control the parameters that characterize the plasma to increase the quality and performance of the industrial processes in which the plasma diagnosis is to be used. The Virtual Instrument can be downloaded for free from a link that is included, in order to be easily adapted to the usual devices in a plasma laboratory.

**Keywords:** plasma diagnosis; Langmuir probe; surface technology

## 1. Introduction

Plasmas are increasingly used in many different industrial processes, such as those related to surface technology, such as plasma-assisted chemical vapor deposition, ion implantation, etching, surface coating, thin films, nanotechnology, etc. [1–7]. Of the many kinds of plasma used in these processes, low-temperature electronic plasmas are the most common. These plasmas present a unique combination of very high chemical activity and high capability for processing at or near room temperature, with the neutral gas temperature, $T_{neutral}$, below 100 °C. Thus, this combination is ideal for processing a wide variety of materials sensitive to temperature, such as semiconductors, polymers, textiles, etc. [8–13]. In the process, the cold plasma modifies only the surface properties of materials, without affecting or degrading the bulk properties. Therefore, plasma can be used to clean a surface, to remove or erode material from the surface, or to deposit a thin film on it. These

cold plasmas are produced by electrical discharges of a direct current or an alternating current, in gases at low pressure (less than 100 Pa). Plasma densities, $n_e \approx n_+$, are obtained in the range between $10^{14}$ and $10^{17}$ m$^{-3}$, $n_e$ and $n_+$ being the electron density and positive ion density, respectively, which must be equal as a consequence of the quasineutrality condition in the plasma bulk. Electron temperature values, $T_e$, are obtained between 0.1 and 10 eV (between $10^3$ and $10^5$ K). The positive ion temperature is similar to that of the neutral gas [7,14–23], since the mean free path of the ion-neutral collision is of the order of 1 mm, this type of collision being the most frequent in this type of discharges and the mean free paths corresponding to the other possible collisions among neutrals, electrons and ions being of the order of magnitude of the characteristic size of the discharge, 0.1 m [3,24].

Plasma diagnosis, consisting in the measurement of the physical magnitudes that characterize the plasma, is essential in the technological processes discussed, as it allows to control these processes to optimize results and enhance its performance.

Among the different plasma diagnosis methods, the method based on the Langmuir electrostatic probes is one of the most used diagnostic methods in all types of plasmas [14,15,21,23,25–37]. Langmuir himself developed the first diagnostic procedures with Langmuir electrostatic probes in 1929. It consists in introducing a cylindrical (although it can also be flat or spherical) conductor or probe into the plasma and measuring its *I-V* characteristic curve, that is, the current collected by the probe as a function of the potential to which it is subjected. This method has several advantages:

- The method allows, from the *I-V* characteristic curve, the diagnosis of multiple quantities characterizing the plasma, such as the following: the potential of the plasma, $V_{plasma}$; the floating potential, $V_{float}$, to which the conductor is polarized when it is in equilibrium with the plasma, so that the net current collected by the probe is zero; the density and temperature of the different species that contain in the plasma, electrons and positive ions, $n_e$, $n_+$, $T_e$, $T_+$; the electron energy distribution function (*EEDF*).
- The method provides different values of several of these magnitudes depending on the zone of the *I-V* characteristic curve and the theory backing the calculations, which allows the results to be compared to ensure their accuracy and quality.
- The Langmuir probe allows performing local measurements, that is, measurements in the immersion zone of the probe, of the parameters described above. This last advantage is very important because it distinguishes this diagnostic method from others that do not provide a local measurement. A drawback of this diagnostic method that could be claimed would be that the probe itself, when immersed in the plasma, causes a disturbance in the plasma. However, as shown in Section 3.3, the ion sheath, which, under certain conditions, forms around the probe, shields the plasma from this perturbation, such that it is negligible as we move away a few millimeters from the probe [7,14–17,22,23,35,36,38].

In this work, different methods for the diagnosis of cold plasmas using Langmuir electrostatic probes are introduced. In general, this method of diagnosis is based on the comparison between the experimental measurements of the *I-V* characteristic curve of a Langmuir electrostatic probe, cylindrical in this case, immersed in the plasma (IVCP) and various theoretical models of the ion sheath surrounding the probe. Indeed, depending on the conditions of the plasma and the zone of the IVCP, there are different theoretical models that allow the experimentalist to correctly diagnose the plasma. These models can be grouped into two groups with two classical limit theories used to describe the intensity of the current collected by the Langmuir electrostatic probe as a function of its polarization potential:

- The orbital limited motion theory, which assumes that charged particles, ions or electrons, fall towards the probe following orbiting trajectories. It can be applied when the mean free paths of the plasma particles involved are long, compared to the scale of the sheath, so that they have little influence on the trajectories, as it is always the case for electrons. It was developed by Mott-Smith and Langmuir in 1920 using conservation laws for the particles in the sheath and it is precise enough to model

the electron current, assuming that the ion temperature is negligible compared to the electron temperature, $\beta = T_+/T_e = 0$ [39]. Bernstein and Rabinowitz extended the theory to take into account the potential profile assuming monoenergetic particles [40]. Finally, Laframboise solved the Bernstein–Rabinowitz model by assuming that the particles follow a Maxwellian distribution function [41].

- The theory of radial movement of the particles towards the probe. It was developed by Allen, Boyd and Reynolds (ABR) for spherical probes [42] and completed by Chen for cylindrical probes to model the positive ion current collected by the probe [43]. It assumes that, after each ion–neutral collision, which, as stated before, is the only type of collision to be considered, the ion loses all its kinetic energy; therefore, after the last collision, it falls towards the probe in a radial movement. Initially, this theory did not consider the thermal movement of the ions, that is, it assumed $\beta = 0$, and was extended by the authors of this article for the case of $\beta \neq 0$ [15,44–50].

It is interesting to note that both classical limit theories, OML and ABR, assume the *EEDF* to be compatible with thermodynamical equilibrium for the electrons and a single electron temperature value, assumption that can be verified given that the *EEDF* can be measured from the IVCP.

The article outline is as follows: in Section 2, the experimental device is described, in which the discharge is produced and on which the measurements described in this article are performed. The IVCP measurement method is also described. In Section 3, the different diagnostic methods used in the different zones of the IVCP are described, as well as the advantages and disadvantages of each of them, and the results that were obtained are compared. Finally, Section 4 comprises the conclusions that can be drawn from this work.

## 2. Experimental Discharge Device and Measurement Method of the IVCP

In this section, a method for measuring the IVCP, that is used throughout the article and that has been widely used by the authors in other works, is presented [14,15,17,18,22,23,36,37,49,51].

Figure 1 illustrates the experimental discharge device and the IVCP measurement device. The discharge was produced in a Pyrex glass cylinder of 31 cm in diameter and 40 cm in height, in which two circular electrodes of 8 cm in diameter were arranged and connected to a very stable source of high voltage (HV) configured as direct current source with an output voltage in the range $V_d(V) \in [0, 2000]$ and an output current in the range $I_d(mA) \in [0, 50]$. The ground was the same for all devices and the metallic lids of the Pyrex glass cylinder, connected to the ground of the laboratory. The pressure of the gas used in the discharge, $p \leq 100$ Pa, was controlled by a mass flow regulator.

### 2.1. Langmuir Probe Design

The manufacture of the probe is an important issue, since a bad design of the probe can lead to bad measurements of the IVCP and, therefore, to erroneous diagnoses of the parameters that characterize the plasma. Figure 2a illustrates a schematic of the probe used in this article. It was formed by a cylinder of tungsten of 6 mm in length and 0.1 mm in radius. Tungsten was chosen to ensure that sputtering or secondary electron emission phenomena that can occur when charged particles impinge on the probe were negligible [7,14,16,34].

Both the size of the probe and the size of its container were carefully chosen so that the required conditions for a good design of the probe were met [22,51,52]. In order for the probe to behave as cylindrical, then $r_p, r_c \ll L$; in order for the thickness of the container sheath to be negligible compared to the size of the probe, then $\lambda_D \ll L$; in order for the probe to be much smaller than the size of the discharge so that the measurement is close to being point-like and does not disturb the discharge appreciably, then $L \ll \lambda_e$. In these inequalities, $r_p$ and $r_c$ are the radius of the probe and of its container, respectively; $L$ is the length of the probe (see Figure 2b); $\lambda_D = \sqrt{\frac{\varepsilon_0 k_B T_e}{e^2 n_e}}$ is the Debye length, which is the characteristic size of the sheath where $e$ is the charge of the electron, $\varepsilon_0$ the vacuum permittivity and $k_B$ the Boltzmann constant; $\lambda_e$ is the mean free path of electron–electron

collisions, comparable to the size of the discharge. It should be noted, in both figures, that, at the end of the container, the probe is guided to avoid touching it, because it would increase the effective surface of the probe, since, over the surface of the container, particles extracted from the electrodes of the discharge, as well as other impurities, may be deposited.

Finally, it is noted that the probe is located in the diffuse afterglow plasma zone where the value of $T_e$ is only one order higher than $T_+$, conditions in which the study of the influence of $\beta = T_+/T_e \neq 0$ is possible, a situation that is important to be taken into account both in the diagnosis and in the technological processes in which this type of plasmas is normally used [7–11,14,18,22,23,49,52].

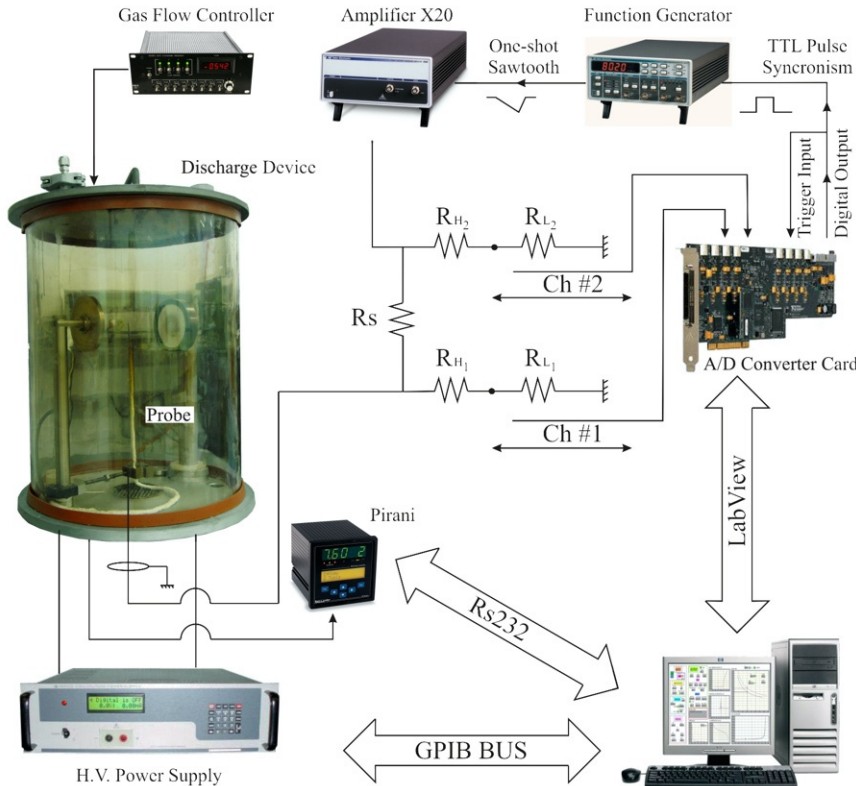

**Figure 1.** Experimental device and measurement setup. Reproduced with permission from Díaz Cabrera et al., Measurement; Reprinted with permission from ref. [14]. Copyright 2014 Elsevier.

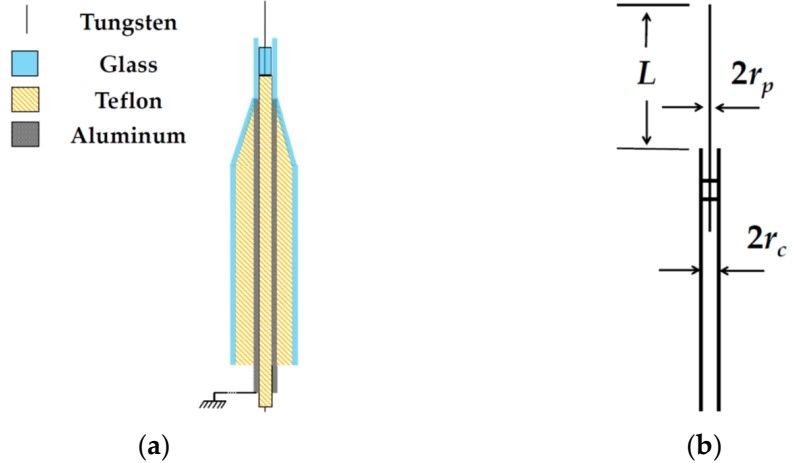

**Figure 2.** (**a**) Probe scheme. (**b**) Relevant dimensions of a Langmuir cylindrical electrostatic probe design.

## 2.2. Analogue Digital Converter Card and Controller CPU

Figure 1 also illustrates the IVCP measurement device. The whole process is controlled by a Virtual Instrument (VI) programmed in the LabView environment [14,23,36]. When the VI is activated, it sends the desired $I_d$ in the HV source and measures the pressure, $p$, of the gas in the discharge. Both variables characterize the discharge conditions in which the IVCP is to be measured. Subsequently, the VI sends the Analogue Digital Converter Card (A/DCC) the command to generate a TTL (Transistor–Transistor Logic) pulse that synchronizes two processes: (1) the function generator emits a sawtooth potential pulse, amplified by the constant gain amplifier and connected to the probe through the resistor $R_S$; (2) the A/DCC begins to measure simultaneously channels *Ch*1 and *Ch*2. However, the polarization values of the probe are usually higher than those allowed by the A/DCC conversion channels of the card (20 V) and that would break the converter card. To adapt these values to the input range of the converter card, a known gain voltage divider, $Gan\# = (1 + R_{H\#}/R_{L\#})$, is placed at the input of each channel. With these measured values, $V_{Ch\#}$, $R_S$, $R_{H\#}$, $R_{L\#}$ and $Gan\#$, the IVCP can be reconstructed by means of the following relationships:

$$V = V_{Ch1} \cdot Gan1, \tag{1}$$

$$I = \frac{V_{Ch1}}{R_{L_1}} + \left( \frac{V_{Ch1} \cdot Gan1 - V_{Ch2} \cdot Gan2}{R_S} \right), \tag{2}$$

where $V$ is the polarization potential of the probe and $I$ is the corresponding current collected by the probe. Regarding this measurement process, it is necessary to make a series of comments:

- In order to increase the measurement accuracy of the IVCP, the measurement of channels *Ch*1 and *Ch*2, $V_{Ch\#}$, should be performed with the greater sensitivity allowed by the A/DCC converter card—16 bit, in our case.
- For the same reason, the A/DCC channels must be carefully calibrated and their zero error must be compensated to ensure maximum sensitivity.
- Likewise, the measurements of channels *Ch*1 and *Ch*2 must be performed simultaneously, since the polarizing potential from the amplified sawtooth pulse varies with time.
- The sawtooth pulse varies from an initial potential of +2 V to the minimum value of −2 V and is amplified by a constant gain amplifier (×20), thereby achieving a linear polarization ramp of the probe between +40 V and −40 V, although the potential pulse sawtooth sweep can be varied to encompass other potential intervals. The number of data measured by both channels, *Ch*1 and *Ch*2, of the A/DCC and its sampling rate must correspond to the width of the constant slope part of the sawtooth pulse. In our case, 2000 data are measured simultaneously by each channel with a sampling rate $f_{sampling} = 5 \cdot 10^5$ samples/s; therefore, the probe polarization ramp must have a duration somewhat greater than $4 \cdot 10^{-3}$ s, so that, during the measurement process, the peak of the sawtooth signal is never reached, which would make the probe polarization nonlinear. This would unnecessarily complicate the data smoothing and data processing algorithm.
- The results using an increasing sawtooth sweep are the same as those obtained using a decreasing sawtooth sweep. However, given that the state of rest between measurements is +40 V, as explained in Section 2.3, the decreasing sawtooth sweep was used.
- Voltage dividers were used to adapt the potential values to those in the input range of the A/DCC, since, as they are passive elements in the circuit, they barely contribute to increasing the noise of the measurement. In our case, $R_{H\#} \approx 1$ MΩ and $R_{L\#} \approx 0.1$ MΩ; therefore, the corresponding gain values were $Gan\# \approx 11$.
- In our case, $R_S \approx 10$ kΩ. These resistance values, $R_{H\#}$, $R_{L\#}$ and $R_S$, must be measured as accurately as possible, because these values are used directly in the formulas to obtain the IVCP.

- As it is shown in Section 3.2, it is essential to minimize the noise inherent in the measurement process, so all cables were coaxial with the shielding of the cables connected to ground, with BNC-type connections. Likewise, all the circuits illustrated in Figure 1 were inside a metal box which was also connected to ground.

### 2.3. Considerations on the Possible Distorsions on the IVCP

The IVCP may be subject to several distortions due to the measurement uncertainty that may arise from the physics of the phenomena involved in the measurement process. The two most important conditions that have to be taken into account in the IVCP measurement are the time involved in the measurement [14,36,51] and the possible contamination of the surface of the probe [53].

It is of utmost importance that the measurement of the IVCP is as fast as possible. Therefore, it should be conducted with the highest sampling rate available, $5 \cdot 10^5$ samples/s, in our case. This importance is due to the fact that the current collected by the probe is very different depending on the zone of the IVCP being measured, with high intensities in the range of high positive polarization potentials; these, as shown in Section 3.1, correspond to the current due to the electrons and low intensity in the range of negative polarization potentials, which correspond to the current due to the positive ions of the discharge. The different magnitudes of the current collected by the probe cause the probe temperature to vary throughout the IVCP measurement process. The Helmholtz potential varies with temperature and constitutes a potential barrier between the probe surface and the sheath [14,21,23,36,38,51,54]; this would imply that this potential barrier would be different in each zone of the IVCP, thus altering the curvature thereof and causing a hysteresis. In other words, the IVCP obtained for an increasing polarizing potential ramp, heating the probe, would not be the same as the IVCP obtained if the polarizing potential sweep was from a high to a low potential, cooling the probe (see Figure 3). As shown in Section 3.2, this alteration of the curvature would cause a great distortion in the measurement the *EEDF*. The way to avoid this problem is to measure very quickly, 4 ms, in our case, so that the probe temperature hardly changes during the measurement process. It should be remembered that the probe is immersed in plasma and neutral gas at low pressure; therefore, the heat exchange flux is very small, the radiation process being the dominating one. Thus, the characteristic times of temperature change of the probe were verified to be greater than 0.1 s, a value much higher than the 4 ms necessary to measure the IVCP. This way, we could ensure that the Helmholtz potential was constant during the measurement and caused a negligible displacement or distortion of the polarization potential of the probe that would have hardly affected the results. As shown in Section 3.2, there is a criterion that ensures the correctness of the measurement regarding this issue.

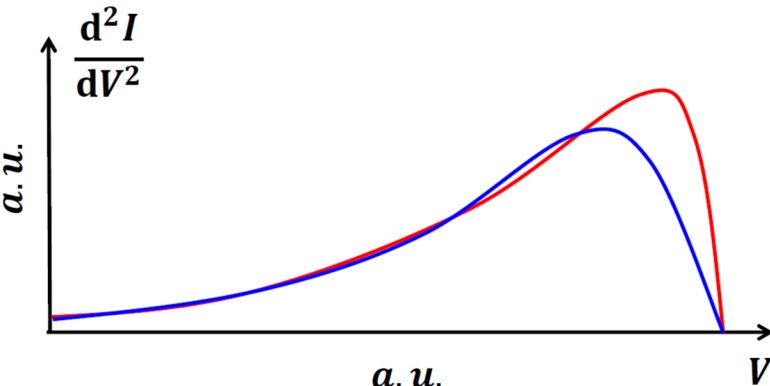

**Figure 3.** The hysteresis in the second derivative of the IVCP, which is related to its curvature, caused by the temperature change of the probe during its measurement process. The red curve corresponds to decreasing potentials, while the blue one corresponds to increasing potentials (authors' experimental measurement).

Hysteresis can also occur, as well as a diminution of the collected current, if the surface of the probe is contaminated with impurities, due to debris removed from the electrodes and other impurities, which also alter the Helmholtz potential on the surface of the probe. This inevitably occurs after a few minutes. Therefore, before performing the IVCP measurement, it is highly recommended to decontaminate the probe. For this reason, it is recommended to polarize the probe with a high potential for a few seconds before carrying out a measurement of the IVCP, so that the high energy electrons impinging on the probe heat it, eliminating such impurities. Therefore, in the state of rest between measurements, our system polarized the probe to +40 V. This implies that the IVCP was measured starting from a high potential using a decreasing ramp. It was verified that the results obtained this way are optimal.

### 2.4. Locality of the Results

As stated in the introduction, one of the advantages of the Langmuir probe diagnosis method is that the measurements are local. The design of the measurement device can be used to better characterize the discharge. Since A/D converter cards normally have several channels—eight, in our case—with the same card, the VI can measure the IVCP of seven different probes, using channels 1–7, each located at a different point on the discharge. The eighth channel would be reserved for measuring the polarization ramp of these probes, as *Ch*2 does in Figure 1, since it is common to all of them. This measurement can be carried out simultaneously in the seven probes or sequentially, according to the requirements that may arise in a particular application. This facilitates, without increasing the cost, the study of plasma uniformity and its spatial variations, which is of utmost importance in large surface technology devices. Likewise, temporal variations or drift processes of the parameters that characterize the plasma can be monitored due to the speed of the entire measurement process, which is less than one second, thanks to the automatization programmed in the VI.

Figure 4 illustrates an IVCP measured using the procedure described. The discharge conditions of this example measurement were Argon Gas, $p = 8.13$ Pa, $I_d = 2.0$ mA. As already stated, the probe was located in the diffuse afterglow plasma zone of the discharge. A summary of the measurement characteristics is included in Table 1.

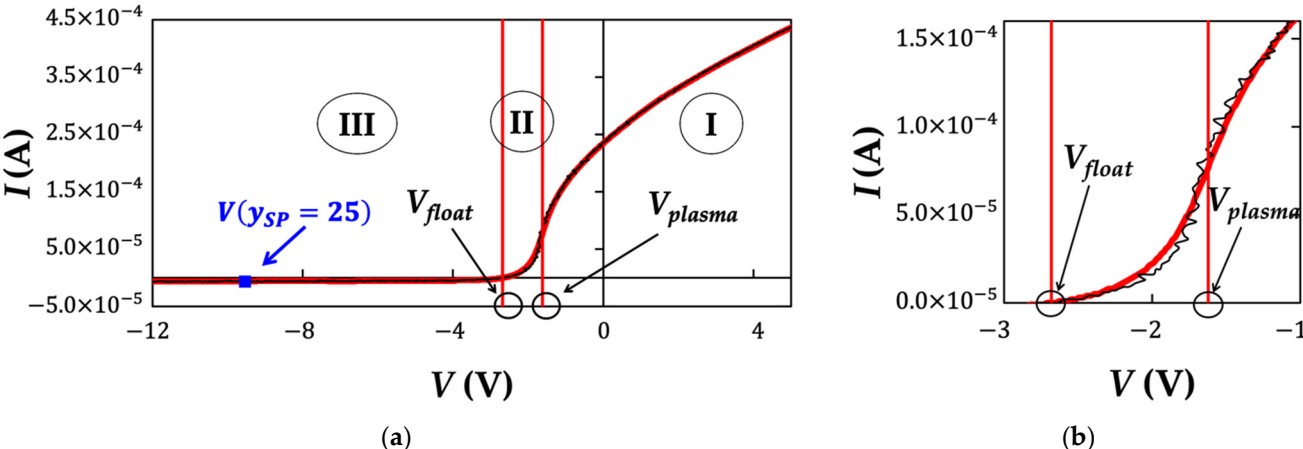

**Figure 4.** IVCP experimental black squares and smoothed, red line; Sonin Plot potential in blue. Discharge conditions: Argon gas; $p = 8.13$ Pa; $I_d = 2.0$ mA; $V_{float} = -2.68$ V; $V_{plasma} = -1.62$ V. (**a**) IVCP. (**b**) Zone of the IVCP for the calculation of the *EEDF*.

**Table 1.** Summary of the characteristics of the measurement process.

| Characteristics of the Measurement Process | |
|---|---|
| Sampling frequency | $5 \times 10^5$ samples/s |
| Number of samples | 2000 samples per channel |
| Duration of the measurement | 4 ms |
| Timestep | 2 µs |
| Average IVCP voltage step in Zone I | 23.8 mV |
| Average IVCP voltage step in Zone II | 12.0 mV |
| Average IVCP voltage step in Zone III | 30.8 mV |
| Sweep starting potential | +40 V |
| Sweep ending potential | −40 V |
| Probe potential between measurements | +40 V |
| A/DCC voltage range | ±5 V |
| Precision of the digital conversion | 16 bits |

## 3. Different Diagnostic Methods Using the IVCP

Once the IVCP has been measured, different diagnosis methods for these plasmas with lower electron temperature can be used on the obtained IVCP, as described in this section. Initially, the simplest case is considered, namely, the plasma is made of electrons and positive ions of a single species, although, since, similarly to the positive ion current, the negative ion current is much smaller than that of the electrons, most of the described diagnostic can also be applied to diagnose plasmas with negative ions, as long as electrons remain the predominant negative species [37,48,55].

Figure 4 also serves to illustrate the scheme of one typical IVCP. As can be observed in the figure, a typical IVCP can be divided into three zones depending on whether the current collected by the probe is mainly due to electrons (*zone I*), positive ions (*zone III*) or a significant mixture of both (*zone II*). In Figure 4, the points corresponding to the floating potential, $V_{float}$, and to the plasma potential, $V_{plasma}$, are indicated.

In the following sections, the different diagnostic methods applied in each of these zones are exposed, along with a discussion on their advantages and disadvantages.

### 3.1. Zone I

Zone I, for values of the polarization potential of the probe greater than $V_{plasma}$, is called the electron saturation zone. It is characterized by the current collected by the probe being fundamentally given by electrons, since, in this situation, these are attracted by the probe, while the positive ions are repelled. This current has the highest values within the IVCP. In this zone, the electrons fall towards the probe following an orbital trajectory without any collision, since, as previously stated, the mean free path of the kinds of collision in which electrons are involved is comparable to the characteristic size of the discharge chamber.

It can be shown that, under these circumstances, the electron current collected by the probe is given by the following equation [14,38,51]:

$$I_e(V) = -A_p n_e e \sqrt{\frac{k_B T_e}{2\pi m_e}} \frac{2}{\sqrt{\pi}} \left( 1 - \frac{e|V - V_{plasma}|}{k_B T_e} \right)^{\frac{1}{2}}, \tag{3}$$

where $m_e$ is the mass of the electron and $A_p$ is the surface of the probe. This expression is valid for values of $V > V_{plasma} + 2k_B T_e/e$, so that the mathematical approximations made are valid [38,51] and to make sure that the current collected is almost exclusively due to electrons.

Different criteria allow us to ensure that electrons are governed by this equation in zone I of the IVCP. First, when the $I^2$-$V$ curve is represented, a straight line should be obtained, as predicted by (3) therefore, the linear correlation coefficient of the linear regression, $r$, should approach unity. Figure 5 illustrates this relationship. Note that this

figure confirms the indicated lower limit, $V > V_{plasma} + 2k_BT_e/e$, since it fails to follow a linear behavior when approaching $V_{plasma}$.

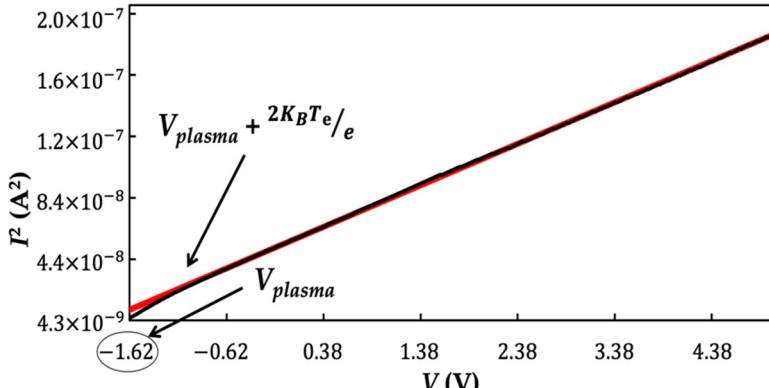

**Figure 5.** $I^2$ versus $V$ curve (black line) of zone I of the experimental IVCP illustrated in Figure 4. The linear zone is constituted by 335 points and fits very well with a linear behavior since the correlation coefficient of the linear regression (red line) is $r = 0.9993$.

Likewise, the criterion of Pilling and Carnegie [7,56] allows us to assure that, in this zone of accelerating electron, the electrons are governed by the OML theory; it consists of the representation of the curve $d(\log_{10} V)/d(\log_{10} I)$. Based on (3), the curve should show a trend towards the value of 2, if the motion of the particles attracted and collected by the probe is orbital. Figure 6 shows this representation for an experimental IVCP. As can be seen, the curve approximates the value of 2, as indicated above.

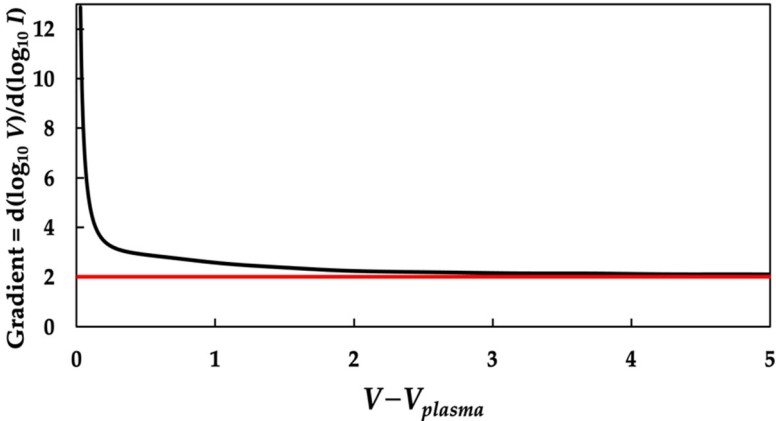

**Figure 6.** Pilling and Carnegie curve of zone I of an experimental IVCP for the same discharge conditions as those in Figure 4.

Thus, the first method of plasma diagnosis is found:

- The slope of the line in the $I^2$ versus $V$ representation allows us to directly obtain $n_e(I^2\text{-}V) \approx n_+$.
- This method has the advantage that it directly determines $n_e(I^2\text{-}V)$ in both electropositive and electronegative plasmas, since the current due to negative ions is very small compared to the electron current, so that its contribution to the zone I current is negligible. Furthermore, a precise knowledge of $V_{plasma}$ is not necessary, since only the linear zone is needed. On the other hand, it should be noted, as a disadvantage, that it can lead to $n_e(I^2\text{-}V)$ results a little lower than those actually found in plasma. This is because the currents collected by the probe are high in magnitude and may produce a depletion in the population of electrons in the space surrounding the probe [14,33].

### 3.2. Zone II

As can be seen in Figure 4, zone II is separated from zone I by $V_{plasma}$ and from zone III by $V_{float}$. In zone II, the current collected by the probe is a mixture of electron current and positive ion current that reach the probe, the current due to electrons being dominant in the points next to $V_{plasma}$ and equal to the positive ion current in $V_{float}$. It is called the electron retarding zone because the potential of the probe is below the potential of the plasma, $V < V_{plasma}$, so that the electrons headed towards the probe are slowed down and only those with a kinetic energy higher than the potential barrier, $V - V_{plasma}$, are capable to reach the probe. Therefore, the electron current decreases when the potential is decreased. Positive ions behave in the opposite way, that is, they are accelerated as the polarization potential of the probe, $V$, is decreased and, when $V = V_{float}$, the positive ion current equals the electron current, so the net current equals zero. In zone II, electrons are in thermal equilibrium within the electric field between the plasma and the probe, so it is assumed that electrons follow a Maxwell–Boltzmann distribution function [7,14,23,37,38,44–49,57]. Thus, the electron current collected by the probe for potential values close to the plasma potential, $V_{plasma}$, follows the following expression:

$$I = I_0 e^{\frac{-e|V - V_{plasma}|}{k_B T_e}}, \tag{4}$$

where $I_0 = I\left(V_{plasma}\right)$ is the value of the current collected by the probe when $V = V_{plasma}$. So, the representation of the log $I$ versus $V$ has a linear behavior, as illustrated in Figure 7, which gives us a method of diagnosis for $T_e$.

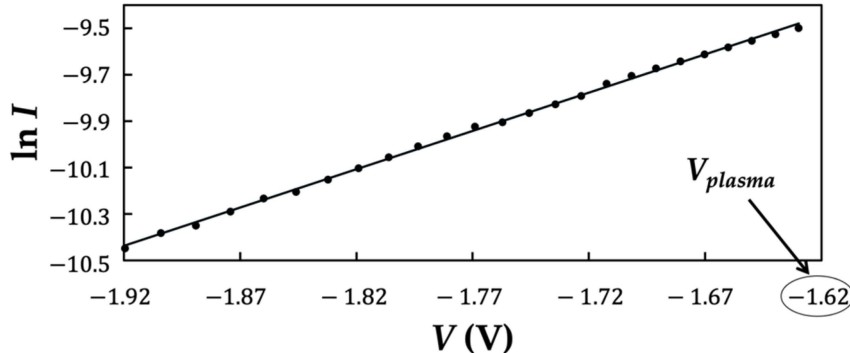

**Figure 7.** Curve ln $I$ versus $V$ in zone II of an experimental IVCP for the same discharge conditions as those in Figure 4. The linear zone contains 25 points and fits very well with a linear behavior since the correlation coefficient of the linear regression is $r = 0.9994$.

The slope of the ln $I$ versus $V$ representation allows us to obtain $T_e(\ln I–V)$. The linearity of such representation ensures the Maxwellian behavior of the electron population in this zone.

Once $T_e(\ln I\text{-}V)$ has been determined, the only physical magnitude that still has to be diagnosed is the value of the plasma potential, to be able to establish the limit between the two zones I and II. For these calculations, $V_{plasma}$ is not necessary, since the methods described for the diagnosis of $n_e(I^2\text{-}V)$ and $T_e(\ln I\text{-}V)$ only require a linear interval in which the slope is calculated. Figure 8 illustrates a comparison between the curves $I^2$ versus $V$ and of ln $I$ versus $V$, of the experimental IVCP from Figure 4. Both linear zones can be observed in the curves; the change in behavior of the electrons, corresponding to the crossing from zone I, where the electrons are accelerated towards the probe, to zone II, where they are slowed down, can be observed. This change in behavior implies the existence of an inflection point in the IVCP, which is where the plasma potential, $V_{plasma}$, is located, separating the two zones.

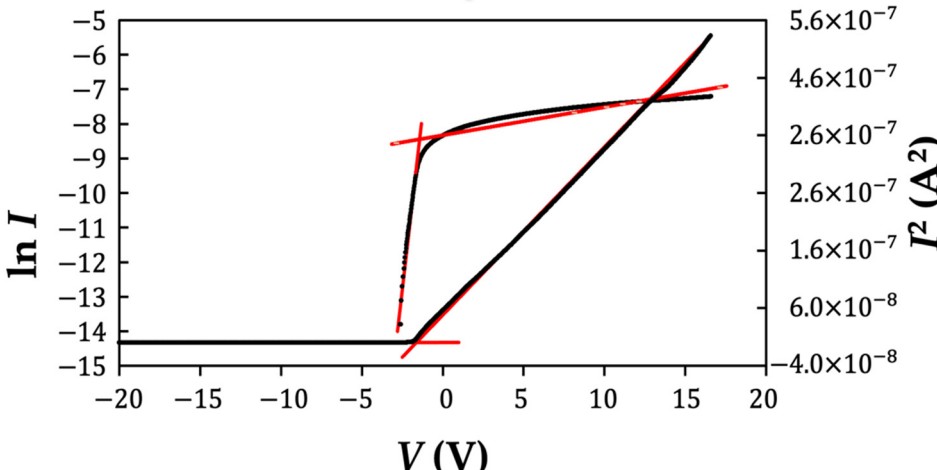

**Figure 8.** Comparison of the $I^2$-$V$ curve and the ln $I$ curve of the experimental IVCP of Figure 4.

However, maximum possible precision in the determination of $V_{plasma}$ is essential for several reasons:

- As previously commented, it separates two zones where the electrons reaching the surface of the probe display different behaviors;
- It allows an alternative method for the determination of $n_e$, which is described at the end of this subsection, given that, when the probe and the plasma have the same potential, the electric field between them is zero and the charged particles reach the probe by means of the mechanism of effusion;
- Finally, $V_{plasma}$ is the reference potential for determining the *EEDF* that provides us with a highly accurate diagnosis method for $T_e$ and $n_e$. Indeed, it was Druyvesteyn, in the 1930s, who demonstrated that the *EEDF* of a plasma can be obtained from the IVCP of a non-concave probe immersed in the plasma, using the following expression [14,23,36,37,51,54,58]:

$$[f_E(E)]_{E=-eV_p} = -\frac{4}{A_p}\sqrt{\frac{-m_e V_p}{2e}}\frac{d^2 I}{dV_p^2}, \qquad V_p \leq 0, \qquad (5)$$

where $V_p$ is the potential of the probe referred to the potential of the plasma, $V_p = V - V_{plasma}$. In this equation, the usual approximation $I = I_e$ is used, since the positive ion current follows an almost linear behavior in zone II, which makes its second derivative negligible compared to the second derivative of the electron current [14,44–46,51]. Given that the numerical second derivative uses three points of the IVCP, zone II of the IVCP should be measured with as many points as possible.

As shown above, to obtain the *EEDF* and a precise value of $V_{plasma}$, the second derivative of the IVCP is needed, which implies a problem; the IVCP is made up of experimental values, so its measurement is accompanied by noise, due to the uncertainty of the measuring process, because a plasma is an inherently noisy system whose noise is amplified by the numerical derivation process dominating the result of the derivation and masking the derivative of the IVCP. Therefore, it is clearly necessary to filter this noise, smoothing out the experimental IVCP before proceeding to the derivation. The methods available for this purpose are diverse, such as, for example, the use of electronic filters and differentiators when measuring the IVCPs and their first and second derivatives [21,54,59]. However, the use of numerical filters on the measured experimental data has given better results. Specifically, the authors proposed a numerical method of smoothing the data from the experimental IVCP [36,51] that has been widely referenced [25,33,60–68]. Firstly, it should be noted that the smoothing process is performed separately on the data measured simultaneously by channels *Ch*1 and *Ch*2 of the A/DCC from which the experimental IVCP is obtained by applying (1) and (2). This is convenient, because the curvature of the line

representing the data from channels *Ch1* and *Ch2* as a function of time is smaller than the curvature of the IVCP, which facilitates smoothing, as shown in Figure 9.

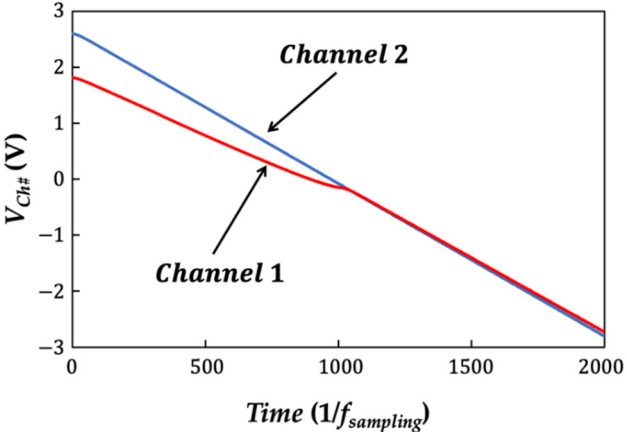

**Figure 9.** Data measured simultaneously by channels *Ch1* and *Ch2* of the A/DCC to obtain the IVCP. The time axis ranges from 0 to 4 ms.

In fact, the data measured by channel *Ch2* correspond to the linear potential ramp polarizing the probe through $R_S$. Therefore, these data are filtered by simply calculating the best linear regression using least squares regression. To ensure the linearity of the set of measured data and the corresponding smoothed linear regression, data are discarded if the correlation coefficient of the linear regression is below a quality threshold, $r < 0.9999$, which would imply a synchronization problem between the triggering of the ramp from the function generator and the sampling of the measurements in the A/DCC. The smoothing of the data measured by the channel *Ch1* is obtained by the iterated convolution of the data with the instrument function of the measurement system, $g(x)$, which is assumed to be Gaussian. The entire iterative process can be performed in a single step by using the convolution product of the data with the function [36,51]

$$g_n(x) = \sum_{k=1}^{n} \binom{n}{k} (-1)^{k+1} \frac{\alpha}{\sqrt{\pi k}} e^{\frac{-\alpha^2 x^2}{k}}, \qquad (6)$$

where n is the number of iterations and $\alpha$ is related to the standard deviation of the Gaussian, $\sigma = 1/\alpha\sqrt{2}$. Typically, it is enough, with $n = 2$ and $\alpha = 15$, to obtain a smooth, noiseless and derivable IVCP. Note that the convolution of channel *Ch1* with the function given by (7) smooths the signal using the neighboring points, so that the lowest $\alpha$ value that allows the numerical derivation should be chosen.

Once the data measured by channels *Ch1* and *Ch2* have been smoothed out, the smoothed experimental IVCP can be obtained by applying (1) and (2) and their numerical first and second derivatives. Figures 4 and 10 illustrate those curves. It can be seen in Figure 4 that the smoothing process was successful since the theoretical and experimental IVCP overlap in the entire measurement interval. Similarly, Figure 10 shows the first and second derivatives, which have a noise-free behavior. Finally, by applying the Druyvesteyn formula (5), the *EEDF* is obtained. Figure 11 illustrates the *EEDF* and the $\ln\left(d^2 I / dV_p^2\right)$ versus $V_p$ curve obtained from the IVCP from Figure 4.

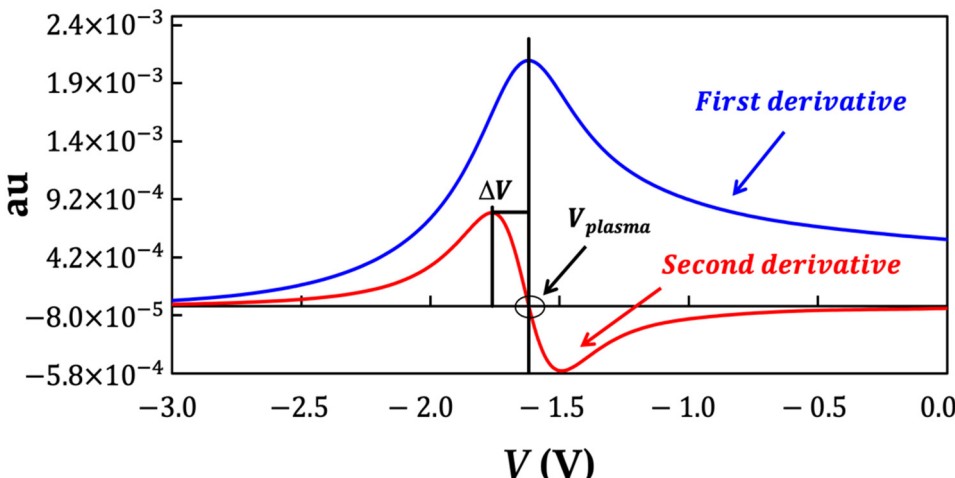

**Figure 10.** First and second derivative of the smoothed experimental IVCP from Figure 4.

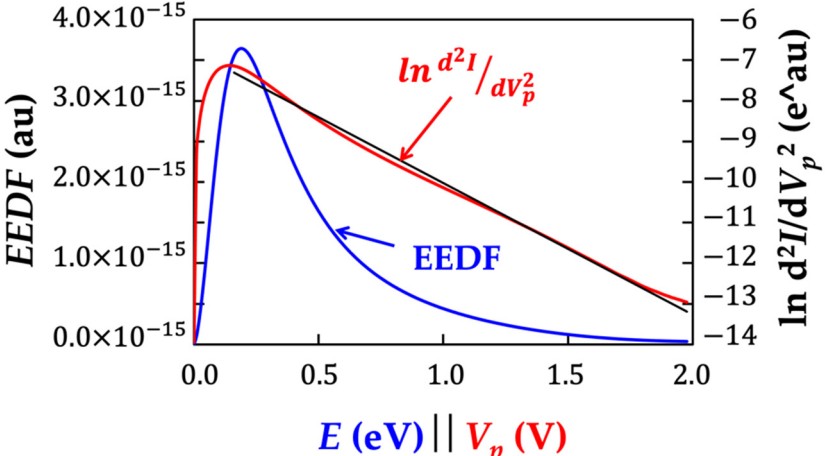

**Figure 11.** *EEDF* and $\ln\left(d^2I/dV_p^2\right)$-$V_p$ curve obtained from the IVCP of Figure 4. Both contain 76 points. The linear zone fits very well with a linear behavior (black straight line) since the correlation coefficient of the linear regression is *r* = 0.998.

These curves are very important because of the diverse information that is obtained from them, explained now in detail as follows:

- The curve $\ln\left(d^2I/dV_p^2\right)$ provides a precise value of $V_{plasma}$, as it corresponds to the inflection point of the IVCP, that is, it is the potential at which $d^2I/dV_p^2 = 0$ (see Figure 10).

- As stated when discussing the required precision for $V_{plasma}$, this plasma potential allows the use of an alternative method for the determination of $n_e\left(I\left(V_{plasma}\right)\right)$, since, when the probe and the plasma have the same potential, the electric field between them is zero and the charged particles and, in particular, the electrons, whose current is dominant at the plasma potential, enter the probe by effusion, so that the value of the current collected by the probe when it is polarized to the plasma potential is

$$I\left(V_{plasma}\right) = I_0 = \frac{eA_p n_e v_{mean}}{4} = eA_p n_e \sqrt{\frac{k_B T_e}{2\pi m_e}}. \qquad (7)$$

This expression, assuming that the electrons follow a Maxwellian *EEDF* [14,36,38,51], once $T_e$ is known by any of the methods described, allows the calculation of $n_e\left(I\left(V_{plasma}\right)\right)$, using a zone of the characteristic different from the previously used to determine

$n_e(I^2\text{-}V)$. It should be noted that $I\left(V_{plasma}\right)$ is taken from the smoothed IVCP to eliminate the uncertainty due to experimental noise.

- The *EEDF* provides another way to determine the electron density of the plasma.

$$n_e(EEDF) = \int_0^\infty f_E(E)dE. \tag{8}$$

This calculation does not rely on the plasma electrons to be governed by a Maxwellian *EEDF*, in contrast with the diagnostic methods for $n_e$ discussed above.

- The *EEDF* also allows us to determine the electron temperature.

$$T_e(EEDF) = \frac{2}{3k_B}E_{mean}, \tag{9}$$

with

$$E_{mean} = \frac{1}{n_e}\int_0^\infty E f_E dE, \tag{10}$$

in the case that the *EEDF* is Maxwellian. If the *EEDF* is not Maxwellian, the effective temperature of the electrons of the plasma, $T_{e,ef}(EEDF)$, is defined as [21]

$$T_{e,ef}(EEDF) = \frac{2}{3k_B}E_{mean}. \tag{11}$$

- The values of $n_e(EEDF)$ and $T_{e,ef}(EEDF)$ obtained using this method are highly accurate and, given that these values do not depend on the premise that the *EEDF* is Maxwellian, these values are used as a reference for comparison with the values obtained by other methods.
- Although, in many cases, the *EEDF* is Maxwellian, it is convenient to check it for each measurement, because, as mentioned, most of the diagnostic methods presented are only applicable for a Maxwellian *EEDF*. In order to compare the results, it is necessary to study the linearity of representation of $\ln\left(d^2I/dV_p^2\right)$ versus $V_p$ given that, when *EEDF* is Maxwellian, the curve plotted should approximate a linear behavior [13,35,36,49,65]. In our case, when the correlation coefficient of the linear regression was below a threshold, $r < 0.9$, it was considered that the *EEDF* was not Maxwellian and the corresponding IVCP was rejected because the described diagnostic methods could not be applied to it. In Figure 11, the straight black line is the linear regression of the curve of $\ln\left(d^2I/dV_p^2\right)$ versus $V_p$, showing that the *EEDF* is Maxwellian.
- Similarly, $d^2I/dV_p^2$ provides a parameter that controls whether the IVCP measurement has been fast enough and the temperature of the probe has not changed during the measurement. As shown in Section 2.3, if the temperature of the probe changes during the measurement, then a variation in the curvature of the measured IVCP takes place; therefore, an alteration of the *EEDF* is found. In general, this change in curvature consists of a displacement of the maximum of the curve versus $V_p$ towards higher energy values, although there are other processes that give rise to the same displacement. Therefore, to control that the displacement is acceptable, within the limits imposed by the measurement process, the following criterion was used: it is acceptable if the absolute value of the potential difference, $|\Delta V|$, which is the difference between the maximum and the inflection point of the curve $d^2I/dV_p^2$ versus $V_p$, fulfills the condition $|\Delta V| < 0.3 E_{mean}$ [14,36,51,54] (see Figure 10). When this criterion is not met, the corresponding IVCP must be discarded. However, a very high value of the standard deviation of the instrument function shown in (6) can also lead to the same displacement, so that, before discarding the measured IVCP, smoothing of the IVCP should be carried out using smaller $\alpha$ values.

- A second control parameter of the smoothing process was also used. In general, the smoothing of experimental data consists in reducing its curvature caused by the additive noise, since the random noise bandwidth is mostly at a higher frequency than the measured IVCP itself. Thus, since the proposed smoothing process is based on the convolution with the instrument function which is assumed to be Gaussian, as the variance of this Gaussian increases, the smoothing increases. Even more, the numerical second derivative uses two additional neighboring points. Thus, if the variance were excessively large, that is, comparable with the variation of the IVCP itself, its curvature could be altered, which would modify and invalidate the results obtained from it, the *EEDF* and the $d^2I/dV_p^2$. To avoid this problem, the following condition is imposed on the variance of the instrument function: $\sigma < 2E_{mean}/3$ [14,36,63].

Finally, once the temperature of the population of electrons has been accurately determined, using the reference value, $T_e = T_e(EEDF)$, the determination of $T_+$ is considered, necessary to obtain $\beta = T_+/T_e$.

- Obtaining $T_+$ is straightforward, since, as stated in the introduction, ion–neutral collisions are, under the usual discharge conditions in this type of processes, those with the lowest mean free path—less than 1 mm—and all other types of collision between charged and neutral particles in cold plasmas have much larger mean free paths, of the order of the size of the discharge. Therefore, it is accepted that the temperature of the positive ions can be considered to be equal to the neutral gas temperature [7,14–23], $T_+ = T_{neutral}$. In this work, a value of $T_+ = 350$ K was used, above room temperature due to the heating from the electrodes that became very hot during operation.

### 3.3. Zone III

This zone is called ion saturation zone, because the current collected by the probe is fundamentally due to positive ions, since $V < V_{float} < V_{plasma}$; therefore, the floating potential, $V_{float}$, separates zone III from zone II. As it is shown in Figure 4, the current in zone III has the lowest value of the entire IVCP, which makes the measurements in this zone difficult to be performed adequately. However, the measurement method proposed in this article achieves precision measurements in this zone of the IVCP, which facilitates the diagnosis of the plasma with minimum disturbance to the plasma. Plasma diagnosis in this zone is optimal for several reasons:

- As mentioned, the current and, therefore, the charge drained by the probe to the plasma in this zone hardly affect the discharge, since, at most, they are of the order of magnitude of $10^{-7}$ A, while the interelectrode current, $I_d$, is, at least, of the order of $10^{-3}$ A;
- Also, the perturbation due to the presence of the polarized probe immersed in the plasma is more efficiently shielded by the positive ion sheath than by the sheath formed in other zones of the IVCP, as measured by the authors [17];
- Finally, plasma diagnosis in zone III is optimal to diagnose $n_+ \approx n_e$, taking into account the temperature of the positive ions, for $\beta = T_+/T_e \neq 0$.

### 3.3.1. The Sonin Plot

The Sonin Plot is used to diagnose in this zone. It consists of a dimensionless representation of the current collected by the probe according to the following equations [7,14,15,22,23,32]:

$$I'\left(x_p, y_{SP}, \beta\right) = \frac{I_+\left(x_p, y_{SP}, \beta\right)}{er_p n_+}\left(\frac{m_+}{2\pi k_B T_e}\right)^{1/2}, \tag{12}$$

$$I'\left(x_p, y_{SP}, \beta\right)x_p^2 = \frac{I_+\left(x_p, y_{SP}, \beta\right)er_p}{\varepsilon_0}\left(\frac{m_+}{2\pi k_B^3 T_e^3}\right)^{1/2}, \tag{13}$$

where $y_{SP} = \frac{-eV_{SP}}{k_B T_e}$, with $V_{SP}$ value of the polarization potential of the probe referred to the potential of the plasma, $V_{plasma}$; $I_+(x_p, y_{SP}, \beta)$ is the intensity corresponding to $V_{SP}$ in the IVCP. This experimental value is absolute, so it must be measured with great precision, hence the considerations made in this regard when describing the IVCP measurement process. Finally, $x_p = \frac{r_p}{\lambda_D}$ is the dimensionless probe radius.

The value of $y_{SP}$ must be chosen carefully, since, on the one hand, it must correspond to values of $V_{SP}$ negative enough so that the current collected by the probe is exclusively due to positive ions; however, it cannot be so negative that there is secondary electron emission on the surface of the probe when the positive ions impinge upon it, as emitted secondary electrons would be confused with positive charge reaching the probe and altering the true value of intensity collected by the probe. To solve these problems, Tungsten was used to manufacture the probe and the value $y_{SP} = 25$ was chosen for the graphs obtained for this article, since it was observed that this value provides the best results, for which $n_+(Sonin, \beta \neq 0) \approx n_e(EEDF)$. The value of $y_{SP}$ can be modified in the VI, if necessary. In Figure 4, a dot is plotted at the potential corresponding to the value of $V(y_{SP} = 25)$.

Figure 12 illustrates the theoretical Sonin Plot obtained for plasmas with positive ions following the ABR theory for $y_{SP} = 25$ and different values of $\beta$ obtained from the authors' model [7,15,18,22,44–50]. In Figure 12, the theoretical Sonin Plot corresponding to the ions following the OML Laframboise theory are also plotted, following the fitting curves by Peterson and Talbot [7,18,22,69].

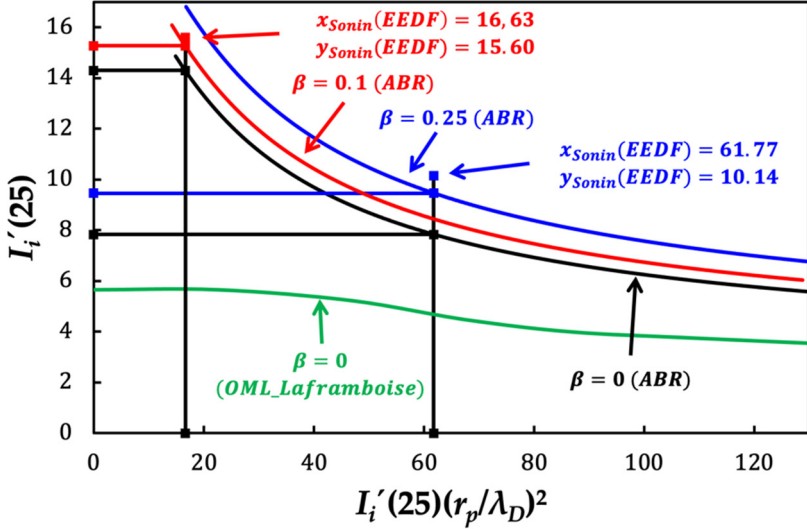

**Figure 12.** Theoretical Sonin Plot for the radial $\beta = 0$ (black), $\beta = 0.1$ (red) and $\beta = 0.25$ (blue) and orbital (green) theories. It also illustrates the cross plotting for $y_{SP} = 25$ used for the calculation of $n_+$.

Plasma diagnosis using this method is performed, firstly, by obtaining the corresponding abscissa of the Sonin Plot using (13). To do this, it is only required to have $T_e$, which can be calculated, for example, from the *EEDF* and $\beta$. Once this abscissa is known, the theoretical radial curve is cross-plotted against this abscissa, for the given $\beta$, from which the corresponding ordinate is obtained; subsequently, using (12), the corresponding $n_+(Sonin, \beta \neq 0)$ value diagnosed from the Sonin Plot in the zone III of the IVCP is obtained. Figure 12 shows the case corresponding to the discharge conditions of the IVCP shown in Figure 4. In this case, the value of $\beta = 0.1$ is small, so that the influence of the thermal movement of the ions is small. This implies that the ordinate of the cross-plotted theoretical Sonin Plot for $\beta = 0$ is very close to the ordinate for $\beta = 0.1$ and the results of the diagnosis for $n_e$ are similar. For this reason, the results from another measurement were added, for which $\beta = 0.25$, for an Argon discharge with pressure $p = 6.84$ Pa and interelectrode current $I_d = 2.5$ mA. In this case, the difference between the ordinates of the

cross-plotted theoretical Sonin Plots for $\beta = 0$ and for $\beta = 0.25$ differ appreciably, resulting in differing values for $n_e$ on account of the upwards displacement of the theoretical Sonin Plot for increasing $\beta$ [44–47]. As a conclusion, the measurement method of the IVCP and this diagnosis method is capable to discern the influence of $\beta$ in the results of the diagnosis.

### 3.3.2. Improvement in the Diagnosis of the Plasma Density

A representation of different values of plasma electron densities diagnosed in different ways, such as the one calculated from the *EEDF*, $n_e(EEDF)$, the one calculated taking into account $T_+$, $n_+(\text{Sonin}, \beta \neq 0)$ and the one calculated assuming cold ions, $n_+(\text{Sonin}, \beta = 0)$ versus the dimensionless ion temperature, $\beta$, is illustrated in Figure 13a and the electron densities diagnosed versus the interelectrode current $I_d$, which is a fundamental magnitude in the characterization of the discharge, are illustrated in Figure 13b. These graphs show that, indeed, for large values of $T_e$, which correspond to small values of $\beta$, the diagnosis does not need to take into account the value of $\beta$. However, as $T_e$ becomes smaller and closer to room temperature, which may occur in several of the technological applications related to the manufacture or treatment of semiconductors, polymers or textiles, the $\beta$ values increase and must be taken into account when diagnosing the plasma. The same would happen if the neutrals were heated, as is the case of PACVD, since it increments $T_+$. The measured $n_e$ and $n_+$ values also diverge when the interelectrode current is increased, given that the increase in the interelectrode current is associated to an increase in $\beta$.

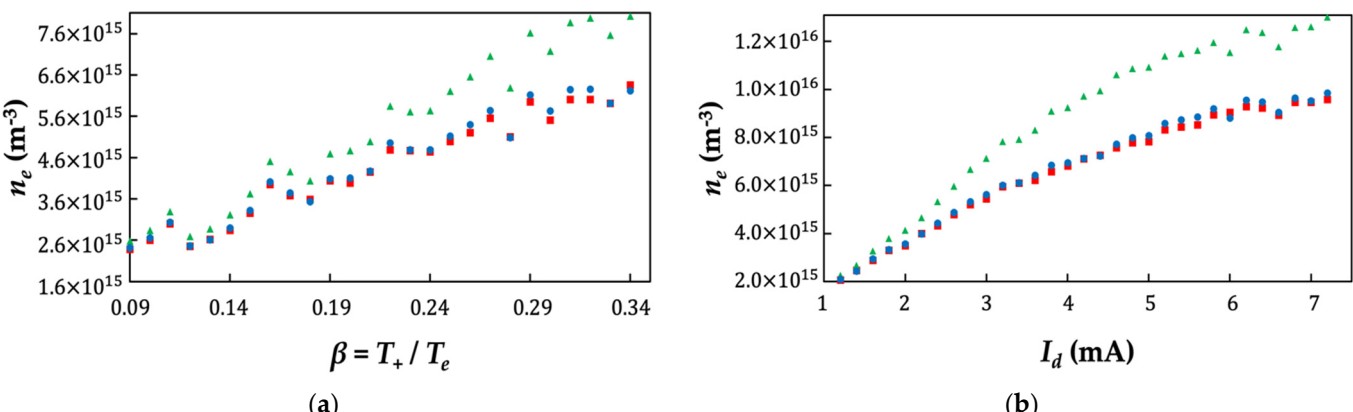

**Figure 13.** (a) Electron density obtained using different methods versus $\beta$. (b) Electron density obtained using different methods versus $I_d$: red squares mean $n_e(EEDF)$, blue circles mean $n_+(\text{Sonin}, \beta \neq 0)$ and green triangles mean $n_+(\text{Sonin}, \beta = 0)$.

It is interesting to note that the quasineutrality condition of the plasma is verified given that the values obtained for $n_+(\text{Sonin}, \beta \neq 0)$ are very close to those obtained for $n_e(EEDF)$, for all $\beta$ values, even if both values have been obtained using different theories applied in different zones of the IVCP.

On the other hand, this method involves a paradox; it is not known a priori which of the two theories, radial or orbital, the plasma ions follow. However, the authors found that, except in very extreme cases [7,18,22,70], the behavior followed by the positive ions in these kind of low temperature plasmas is systematically radial. Indeed, only for Helium plasmas with low plasma density the orbital theory was found to be valid [7,18,22,70]. In any case, a quick estimate can always be made using the values of $n_e$ and $T_e$ obtained with any of the aforementioned methods and comparing the results. Figure 14 illustrates that, indeed, the OML behavior can be ruled out, since the points obtained with the *EEDF* results are very close to and above the curve corresponding to the radial behavior, for $\beta \neq 0$.

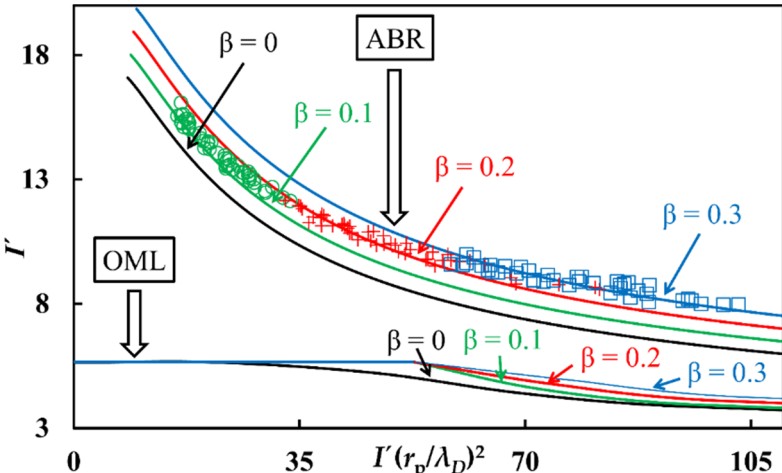

**Figure 14.** Sonin plot including the experimental data (symbols) and theoretical curves (solid lines) for argon plasmas. Circles mean $0.08 \leq \beta \leq 0.15$, crosses mean $0.16 \leq \beta \leq 0.25$ and squares mean $0.26 \leq \beta \leq 0.34$. Reproduced with permission from Díaz Cabrera et al., Plasma Sources Science and Technology; Reprinted with permission from ref. [18]. Copyright 2015 IoP.

Figure 14 illustrates the theoretical Sonin Plots versus the experimental points from the EEDFs for 153 different Argon discharge measurements under different conditions of pressure and interelectrode current. The series were taken with the interelectrode current ranging from 1.2 mA to 10 mA in 0.5 mA steps until the HV power source voltage limitation was found. Similarly, the pressure ranged from 2 Pa to 10 Pa in 0.3 Pa steps. As can be seen, for all the points obtained from the *EEDF*, the location of the points is always in accordance with the curves predicted by the radial theories. Furthermore, the points follow the evolution of the parameter $\beta$. This further supports the precision of the method. The OML curves are not consistent with any of the measurement performed with Argon in the conditions of the experiments. It is interesting to note that the OML model also predicts an increase in the ion current collected when the ion temperature is taken into account, as shown in Figure 14, following the fitting curves by Peterson and Talbot [7,18,22,69]. More advanced orbital models, common in other cold plasma studies, such as space plasmas, studied that consider the thin sheath [71] or the collisions in the sheath [72,73], that also predict a higher ion current in order to find closer agreement with measurements in these or similar conditions should be used.

This method can also be adapted to the diagnosis of negative species, since our models can be used in the case of several negative species of any kind provided that the positive ions approach the probe following the radial theories. Therefore, it is possible to obtain the theoretical Sonin Plot to diagnose the plasma and cross-plot the density of positive ions, $n_+(\text{Sonin}, \beta \neq 0)$. Franklin and Snell [55] showed how to obtain an appropriate value for the temperature of negative ions, $T_-$, depending on the conditions of the plasma. From the *EEDF*, $T_e$ and $n_e$ can be obtained. Finally, the quasineutrality condition allows us to establish $n_- = n_+ - n_e$ [37,44–48,50,55,57,74,75].

In Table 2, a comparison is shown between the values of $n_e \approx n_+$ and $T_e$ obtained by means of the different methods presented in this work, both for the discharge for which the IVCP of Figure 4 was obtained, with $\beta = 0.1$, and for the case included in Figure 11, with $\beta = 0.25$. The chosen $\beta$ values are examples of the two extremes in the electron temperature in the series of measurements included in this work and serve to illustrate the possible differences that arise in the plasma diagnosis when the ion temperature is not negligible compared to the electron temperature. It is shown that there is good agreement between the values obtained from the density of the *EEDF* and the Sonin plot for $\beta \neq 0$, while the value for $\beta = 0$ slightly deviates to higher values, since it corresponds with lower ordinate values in the cross-plotting, this difference being greater for the case with $\beta = 0.25$. Finally,

the values obtained from $I\left(V_{plasma}\right)$ and $I^2$-$V$ area somewhat smaller, since, as mentioned, the current collected by the probe in this zone are important and a depletion of electrons and a decrease in the electron density may occur in the space around the probe.

**Table 2.** Comparison of the results obtained using the different plasma diagnosis methods as a function of $\beta$.

| | | | | Plasma Diagnosis Method | | | |
|---|---|---|---|---|---|---|---|
| $\beta$ | $n_e$(*EEDF*) (m$^{-3}$) | $n_+$(*Sonin*,$\beta \neq 0$) (m$^{-3}$) | $n_+$(*Sonin*,$\beta = 0$) (m$^{-3}$) | $n_e$($I^2$-$V$) (m$^{-3}$) | $n_e$($I(V_{plasma})$) (m$^{-3}$) | $T_e$(ln$I$-$V$) (K) | $T_e$(*EEDF*) (K) |
| 0.1 | $2.5993 \times 10^{15}$ | $2.6515 \times 10^{15}$ | $2.8436 \times 10^{15}$ | $2.2422 \times 10^{15}$ | $1.8016 \times 10^{15}$ | 3544 | 3681 |
| 0.25 | $4.9923 \times 10^{15}$ | $5.1232 \times 10^{15}$ | $6.2038 \times 10^{15}$ | $2.8716 \times 10^{15}$ | $3.1560 \times 10^{15}$ | 1362 | 1360 |

## 4. Conclusions

In this article, different methods of plasma diagnosis using Langmuir electrostatic probes are described, suitable for the case of plasmas with electron temperature close to room temperature. The following conclusions should be highlighted:

- The method here described for measuring the current-to-probe voltage characteristic curve of a Langmuir probe immersed in a low temperature plasma, or IVCP, complies with the strict precision requirements needed to diagnose the plasma using any of the methods described, including the method that uses the ion saturation zone, which is particularly difficult because the current collected by the probe is very small.
- Likewise, the necessary qualities of a Langmuir electrostatic probe are described, as well as the conditions that its size, shape and container must meet.
- The measurement of the IVCP allows to diagnose the plasma locally, which implies measuring magnitudes that characterize the plasma in the region where the probe is located. This makes it possible to monitor and control the discharge to improve the quality and performance of these industrial processes, such as repeatability, by exploring the discharge parameters that provide better results; uniformity, by allowing the measurement at various points at the same time and for the same cost; or time drift, by processing the measurement and data treatment in a very short time, less than a second.
- Depending on the zone of the characteristic, different diagnostic methods are here detailed, their results are compared and each one of their advantages and disadvantages are weighed. These methods are based on different sheath theories in plasmas that are referenced. The values of the different plasma parameters obtained in the different zones of the IVCP are consistent with the theories, with remarkably good agreement between the results obtained using the measured *EEDF* and the ion saturation radial model taking into account the ion temperature.
- The entire process of measuring the IVCP and data processing to obtain the physical magnitudes that characterize the plasma by different methods was automated by programming a Virtual Instrument, VI, in the LabView environment. The entire data acquisition and data processing takes less than a second. The VI can be downloaded for free. Adapting this VI to the devices from other plasma laboratories is relatively straightforward.

**Author Contributions:** Conceptualization, G.F.R., J.M.D.-C. and J.B.; methodology, J.M.D.-C. and J.B.; software, G.F.R. and J.M.D.-C.; validation, J.I.F.P. and J.B.; formal analysis, J.I.F.P. and J.B.; investigation, G.F.R. and J.M.D.-C.; resources, J.I.F.P. and J.B.; data curation, G.F.R. and J.M.D.-C.; writing—original draft preparation, J.B.; writing—review and editing, G.F.R., J.M.D.-C., J.I.F.P. and J.B.; visualization, J.I.F.P. and J.B.; supervision, J.I.F.P. and J.B.; project administration, J.M.D.-C.; funding acquisition, G.F.R. and J.I.F.P. All authors have read and agreed to the published version of the manuscript.

**Funding:** This study was partially supported by the Spanish Ministry of Science and Innovation, Ref. No. FIS2010-19951, which is partially financed with FEDER funds and the FPU Program of the Spanish Ministry of Education (Ref. FPU17/01387).

**Institutional Review Board Statement:** Not applicable.

**Informed Consent Statement:** Not applicable.

**Data Availability Statement:** Data is contained within the article.

**Conflicts of Interest:** The authors declare no conflict of interest.

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
