# Peer review of "Low Electron Temperature Plasma Diagnosis: Revisiting Langmuir Electrostatic Probes"

_coatings, doi:10.3390/coatings11101158_

Round 1

Reviewer 1 Report

This manuscript described a method of measurement of the current to probe the voltage characteristic curve of a Langmuir electrostatic probe immersed in a plasma characterized by a low electron temperature.  This manuscript is interesting, however, there are some issues should be addressed. Meanwhile, the organization of this article can be significantly improved before it can be accepted for publication. Here are some main suggestions listed below.

1. The main finding or scientific improvement should be addressed in the section of abstract.

2. The expression of bullet point of literature survey and discussion in the section of Introduction should not be used too much. The structure in this article can be improved.

3. Some figures can be combined, for example, Figs. 13 and 14. 

4. The resolution of Fig. 1 is too low. 

Here are some comments and the author can go through all the manucript to reorganizing this article.  

Reviewer 2 Report

The presented manuscript titled “Low electron temperature plasma diagnosis: Revisiting Langmuir electrostatic probes” is a continuation of extensive work by the authors. It is generally well structured and scientifically sound but requires minor corrections before publishing.

I like and support the act of sharing the LabView Virtual Instrument very much.

The first half of the manuscript needs moderate grammar editing, Demonstrative (this, that) are used too often and make the text hard to understand for the reader. Example in lines 69-71, 241…

Line 40-43, run-on sentence. There is one “and” too many in line 41. Electron temperature should be explained in a separate sentence.

Line 52-53, “one” repeats unnecessarily.

Lines 109-113, 117, 224, 229… Passive should be used. Instead of “we will describe”, “it will be described”, as it is more common for research articles.

Line 133, I suggest either “erroneous results” or “erroneous diagnosis”, not both.

Line 163, abbreviation TTL should be explained.

Line 241, the word “values” seems unnecessary to me.

Line 245, optimal instead of optimum.

Line 247, “is” is redundant.

Line 258, what does “likewise” refer to?

Line 270, “up” is redundant in my opinion.

Line 506, Can a temperature of 350 K be called room temperature? Or just close-to room temperature?

Line 625, typo

Different points/bullets are used throughout the manuscript, consider unifying them.

Some parts of the conclusions sound like abstract.

Reviewer 3 Report

The paper describes the measurement of I-V curves from a cylindrical Langmuir probe in a laboratory-based discharge plasma. The paper summarizes relevant theories related to analysis of Langmuir Probes. The paper is well-reasoned and should be very informative for the readers. There are however some issues which must be addressed before the paper is considered for publication. Some major and minor comments are presented. The minor comments can be addressed easily while some of the major comments, if addressed, would significantly increase the impact of the paper. One key aspect that would also improve the paper and aid the readers is to divide section 3 into smaller subsections, which can be then easily referenced in some of the earlier text. Minor comment 4 is about this issue; so it is encouraged that the authors add subsections and more logical breaks.

Major Comments:

1. Table 1 and associated methods of getting ne and Te: The authors present different methods to obtain Ne and Te estimates. Why are only two Ti to Te ratios used here? What are the main outcomes from this table? What are the range of beta values that can be obtained using the plasma source ? The authors need to show some more data points or at least convince the readers that these two instances are enough to draw some conclusions.

Also, there are several methods that have been used in the past to analyze Langmuir probes. For example, Barjatya et al (2009) argued that the OML saturation current is not always the ideal thick sheath estimates and thus they can be approximated with a parameter beta (not to be confused with the authors’ present usage of beta as Ti/Te). The ‘beta’ exponent can vary substantially from 0.5 as is expected for cylinder depending on the plasma. In the same data analysis method, it is shown that a curve-fitting approach yields an accurate Ne and Te for plasma encountered in orbit.  While orbital plasmas are mesothermal, it is still interesting to see the results from the curve-fitting approach to the laboratory I-V curve with appropriate equations. How do the Ne, Vplasma and Te estimates compare for this method versus the ones they use already? Link to the paper: https://doi.org/10.1063/1.3116085. In short, it would be useful to add some columns like Ne(Curve-fit) and Te (Curve-fit) that can use the analysis method presented in the aforementioned reference.

2. Upsweep vs downsweep: The narrative in page 6 (lines 204-50) is interesting and relevant. The authors describe the importance of sampling rate in the light of hysteresis. Hysteresis is also produced by contamination (Piel et al., 2001 https://doi.org/10.1088/0022-3727/34/17/311) and the authors state that the probe is cleaned at high positive potentials. Is this true before every I-V measurement? Are the sweeps upsweep or downsweep ? It would be useful to show that the hysteresis or contamination effects are absent with used cleaning and sample rates in the I-V curve used as they might affect Te. Also, before presenting Fig 4, I would suggest creating a table to describe all the characteristics of the I-V sweep e.g. step-size of the voltage applied ? How many points per sweep ? Nature of sweep (Up/Down) ? Sampling time ? Some of these are written already but they are not arranged in a way to help the reader get all the information quickly.

3. EEDF: The EEDF depends, to some degree, on the voltage step and also the subtraction of the ion current. Equation 5 is referenced to the total I-V curve but in reality some portion of this is the ion current. Have the authors considered subtracting the ion current ? If so, how ? Does that affect the Te and Ne measurements through EEDF ? The numerical derivative is also sensitive to the voltage step all of which need to be discussed or commented.

Minor Comments:

1. Fig 15: The authors mention that the data is generated from 153 different Argon discharge measurements, but the statistics of these measurements are not presented. What are the discharge currents, pressures and Te that are covered with these discharge measurements? The authors say that the data support the radial theory except in extreme cases and they refer the relevant papers, but they should mention here a summary of these cases to provide context especially since the paper is a review of applicable methods.

2. The effect of smoothing the I-V curve: The authors describe a smoothing algorithm with citations around lines 380. They also correctly state that this influences the derivatives. However, this effect has not been shown in a plot or on the I-V curve. For example, in Table 1 the Ne(I(V-plasma)) is different from the other Ne estimates. Does the smoothing change the first or second derivatives enough to change the V-plasma? Also, the authors need to show the smoothed I-V curve with zoomed in focus of the retardation region for Fig 4 since the Te can be affected by this. In its current form it is hard to discern how good the smoothing is because of the pixels in the plot.

3. What is the ground of the whole setup? Chamber walls? The Vplasma is negative so does that mean there is positive charging going on? What is the floating potential for the I-V shown in Fig 4?

4. The authors have referred in some sections to statements such as “As we will see” (Example line 246). It would help the readers if these are referenced to certain sections. The narrative lacks logical breaks such as subsections that would help this referencing. It is difficult to look for the section without these references. Similarly, the use of Sonin coordinate in Fig 4 in blue is not mentioned in caption which may puzzle the uninitiated reader.

5. Typographic error around Line 625.

6. Line 646. ‘Textiles…’: Use a formal end to the sentence.

7. Line 34: Give a range to what cold plasmas would refer to in terms of eV or K to help the reader.

Round 2

Reviewer 1 Report

The authors addressed my comments. I suggest this manuscript can be accepted for publication.

Author Response

Many thanks to the reviewer for their contributions to the work, which have helped to improve the work.

Reviewer 3 Report

The authors have done a commendable job addressing the revisions. There are some clarifications that need to be made however before the paper is considered for publication.

  1. The authors have pointed out that in response to Major comment 1 of revision 1 that the main motive of Table 1 (Table 2 in revised manuscript) is to highlight the impact of the temperature ratio. They show that ne(EEDF) and n+(Sonin,β≠0) are in close agreement. This seems to be one of the significant takeaways but this is not sufficiently highlighted in the conclusions. The conclusions presently have no key highlight and statements such as ‘A good agreement has been found between the results obtained using the different zones and the different theoretical models’ – need to be more precise. I suggest the authors highlight their key contributions through a list in the concluding paragraph highlighting the key points.
  2. In response to Major comment 1 second part, the authors cite a lack of time to complete this part. However this method is a major technique in relation to Langmuir probe analysis. While the issue of time can be agreed on, it is perhaps incorrect to classify this method as a simple OML equation since unlike the thick sheath assumption, the method works for thin sheath as well as utilizing the fitting of the current increase in the saturation region. The authors should at least mention this method in the literature in reference to cold temperature space plasma data analysis.
  3. I appreciate the inclusion of Table 1 but I do not find any reference of why a downsweep was chosen (+40V to -40V) in contrast to an upsweep ? This should have some impact if the contamination removal was not successful. The authors should clarify.

I am happy with the remaining changes.

Author Response

This manuscript is a resubmission of an earlier submission. The following is a list of the peer review reports and author responses from that submission.